# Response of a Radiology Department to the SARS-CoV-2 Pandemic: The Experience of the Hospital “Policlinico Tor Vergata” in Rome

**DOI:** 10.3390/ijerph18105255

**Published:** 2021-05-14

**Authors:** Andrea Malizia, Laura Filograna, Francesco Paolo Sbordone, Giorgio Ciccarese, Andrea Carbone, Beatrice Carreri, Colleen Patricia Ryan, Gian Marco Ludovici, Andrea Chierici, Guglielmo Manenti

**Affiliations:** 1Department of Biomedicine and Prevention, University of Rome Tor Vergata, Via di Montpellier 1, 00133 Rome, Italy; malizia@ing.uniroma2.it (A.M.); manenti@med.uniroma2.it (G.M.); 2Policlinico Tor Vergata: Fondazione PTV, Via di Montpellier 1, 00133 Rome, Italy; laura.filograna@ptvonline.it (L.F.); francescopaolo.sbordone@students.uniroma2.eu (F.P.S.); giorgiociccarese@gmail.com (G.C.); andrea.carbone.20@students.uniroma2.eu (A.C.); beatrice.carreri@studens.uniroma2.eu (B.C.); c.ryan@hsantalucia.it (C.P.R.); 3Department of Industrial Engineering, University of Rome Tor Vergata, Via del Politecnico 1, 00133 Rome, Italy; a.chierici@studenti.unipi.it; 4School of Engineering, University of Pisa Largo Lucio Lazzarino, 2, 56122 Pisa, Italy

**Keywords:** SARS-CoV-2, COVID-19, emergency management procedures, radiology

## Abstract

The dissemination of severe acute respiratory syndrome linked to the novel coronavirus, SARS-CoV-2, prompted all health services to provide adequate measures to limit new cases that could affect healthcare professionals. Due to the large number of suspected patients subjected to CT scans and the proximity of radiologists to the patient during exams, radiologists as well as the entire staff of the radiology department are particularly exposed to SARS-CoV-2. This article includes the emergency management procedures, the use of personal protective devices, and the rearrangement of exam rooms and of human resources in the department of radiology at “Policlinico Tor Vergata” in Rome performed during the SARS-CoV-2 pandemic. We introduce the management measures that our department has taken to cope with the influx of patients while still ensuring the proper management of other emergencies and time-sensitive exams.

## 1. Introduction

In December 2019, a new coronavirus disease, COVID-19, was reported in China. Within 3 months, the World Health Organization defined COVID-19 as a pandemic, with more than 370,000 cases and 16,000 deaths worldwide. Contamination occurs easily, mainly through droplet and contact transmission. The highest risk of transmission happens within 91 cm, but respiratory droplets from infected patients can travel up to 183 cm from their source [1].

This is why prevention planning and control measures of the transmission of the virus to patients and department staff members is fundamental in each health department, especially in the radiology department. Furthermore, the transmission speed and the incidence of the pandemic make it necessary to optimize the use of personal protective devices and dedicated COVID-19 equipment, as the availability of supplies is limited [2].

In February 2020, “Diagnosis and Treatment Protocols of COVID-19 Infection” was published by the National Health Commission (NHC) of China. It confirmed that radiology is one of the frontline specialties in the clinical service, as diagnostic imaging exams, such as X-ray and computed tomography (CT) examinations, are effective methods for the screening, diagnosis and follow-up of this infectious disease [3,4]. 

A high sensitivity, fast scanning speed, and high spatial resolution for disease diagnosis are some of the advantages of a CT scan compared to real-time reverse transcriptase polymerase chain reaction (RT-PCR) for viral nucleic acid, which still remains the reference standard for COVID-19 diagnosis. Chest CT scans can show early signs in the lungs of COVID-19 infection, including abnormalities of the interstitium and lung parenchyma, and help to assess the severity of the disease [4]. Moreover, some studies have recently reported that a CT scan can outperform molecular testing as it is a more sensitive test for the virus. This is why, in China, CT is now used as the main imaging method for screening and diagnosis of coronavirus disease [5]. The role of digital chest radiography (DR) in COVID-19 is very limited, especially for detecting subtle and early inflammatory changes in the lungs (especially pulmonary interstitial changes), which determine many missed diagnoses. However, for critically ill patients who cannot be moved to the CT room (especially in case of tracheal intubation and mechanical ventilation), the medical doctor can choose mobile DR to evaluate the severity of lung lesions [3,4,5].

With the aim to suggest measures for reducing risk of infection of medical staff and patients subjected to radiological examinations, in the present paper we describe the experience and the measures adopted at the Radiology department of “Tor Vergata” university hospital related to practical, staff protection, logistic, planning, patient management and educational issues during the first peak of SARS-CoV-2 pandemic.

## 2. Remodulation of Hospital

“Policlinico Tor Vergata” (PTV) is a university medical center associated with the University “Tor Vergata”. In March 2020, this hospital was designated to care for patients with COVID-19. Three areas have been created. The Red Areas (Hot Zones), which are the areas with the highest risk of contamination, the Yellow Areas (Warm Zones), which are the areas with a moderate risk of contamination, and the Green Areas (Cold Zones), which are the areas with the lowest risk of contamination.

Consequentially, during the outbreak stage, “Policlinico Tor Vergata” adopted a series of prevention and control measures. The first inpatient building, located in the emergency department, was reconfigured to handle cases of COVID-19. Fever tents were set up by the emergency department to separate normal emergency patients from patients with symptoms or exposure history suspicious of COVID-19. The hospital was divided into two areas: the contaminated area and COVID-free area. The contaminated area is designed to host patients suspected of harboring the virus and confirmed patients, who are placed in an isolation ward; the COVID-free area is instead designed to host patients who are not suspected of harboring the virus. The suspicion of SARS-CoV-2 infection is based on the signs and symptoms of the disease and the exposure history suspicious of COVID-19 [6].

## 3. Reconfiguration of the Radiology Department

In this context, our radiology department underwent organizational and structural changes according to the hospital reconfiguration.

In a non-pandemic period, the radiology department is divided into two sections:

The emergency radiology department located within the emergency department. The emergency radiology department evaluates and carries out the diagnostic examinations requested by the emergency department. Interventional radiology encompasses intra- and extra-vascular emergency operations and planned operations. The equipment of the emergency radiology department includes 1 magnetic resonance (MR) scanner, 1 computed tomography (CT) scanner, 1 set of digital radiography (DR) systems and 1 mobile bedside drive medical (DR) set, 1 ultra sound (US) scanner and 5 imaging diagnostic workstations for picture archiving and communication sustem (PACS).The central radiology department is divided into diagnostic and interventional radiology sectors. The central radiology department uses the rest of the equipment including US, DR, CT and magnetic resonance imaging MRI to perform emergency examinations. The planned examinations required by the medical and surgical departments for hospitalized patients and planned examinations for outpatients are performed there as well. The equipment of the central radiology department includes 3 MR scanners, 4 CT scanners, 2 digital subtraction angiography (DSA) systems, 6 sets of digital radiography (DR) systems and 1 mobile bedside DR set, 7 US scanners, and 22 imaging diagnostic workstations for PACS.

During the outbreak, the radiology department was reconfigured. 

The central radiology department became a COVID-free area, and the emergency radiology department became a contaminated area. Thus, the Central Radiology Department now carries out the emergency examinations and planned examinations required by the medical and surgical departments of the COVID-free area for hospitalized patients as well as the planned examinations for outpatients and the examinations required by the COVID-free area of the emergency department. The emergency radiology department carries out the diagnostic examinations requested by the emergency department of the contaminated area and by the medical and surgical departments of the contaminated area for hospitalized patients.

The main problem of this organization regards patients with suspicious symptoms, who may be not infected, and who otherwise can become infected in the contaminated area. To overcome this problem, the rules for sanitizing of the contaminated area must be respected.

### 3.1. Impact of the Infection among the Staff Working in the Radiology Department

The Radiology Department suffered an impact during the second wave of the pandemic that took place in Italy mainly between mid-October and mid-November 2020, as is shown in Table 1.

The impact of the temporary loss of staff due to infection deserves some considerations. Among the 5/33 radiologists infected, up to 4 were absent from work. However, the impact on the radiology department activity was low due to the scant number of radiologists infected and the reduction in the workload in the radiology department related to cancellation of diagnostic examinations of non-oncological or non-suspected-oncological or non-urgent outpatients. The quality of healthcare provided was not affected at all.

Likewise, the infection of technicians (4/60) and nurses (5/13) had no impact on the activity of the radiology department for the same reasons. Furthermore, the headcount of these professionals, particularly the former, was increased at the beginning of the pandemic.

The low number of resident doctors affected (5/48) and the peculiarity of their role within the radiology department were equally responsible for the scant impact on the department activity of resident infections.

### 3.2. Outpatient Management

During the first peak of the pandemic, the planned diagnostic examinations of outpatients were cancelled except for oncological patients, suspected oncological patients and for urgent patients.

During the first wave and today, every outpatient, after the fever checkpoint, underwent a specific questionnaire regarding signs, symptoms and exposure history suspicious of COVID-19. To access the diagnostic examinations, the outpatients must complete a questionnaire regarding the following signs and symptoms: fever, cough, malaise, exhaustion, fatigue, headache, chills and muscle aches. The suspicion of exposure history of COVID-19 includes exposure risk in the last 14 days and travel to risk areas in the last 14 days. A single positive answer to the questionnaire prevents the outpatient accessing the planned diagnostic examination.

### 3.3. Management of Patients in the Radiology Emergency Department 

When a non-COVID patient comes to hospital for imaging exams, his or her temperature is measured, and epidemiological history, symptoms, and signs are taken by a member of the staff. If there are suspected criteria or the temperature is higher than 37.2 °C, the patient is brought to the fever tent for further screening. Patients with no exposure history, suspicious symptoms, or fever are screened in one of the non-contaminated CT scanners. The technicians assigned to these scanners, as well as all patients and the people accompanying them, are required to wear surgical masks. After the CT examination, the technician analyzed the images quickly and, if suggestive of lung infection, they immediately reported to the radiologist on duty. If COVID-19 infection is excluded, the patient can leave the CT examination room while the medical doctor studies the lung infection. The patient is then immediately reported and sent to the fever tent. The floor, equipment and air in the CT examination room are then disinfected before examining other patients according to regulations. These CT scanners are considered to be not contaminated (not fever-CTs) after these sanitization procedures [7].

## 4. Various Safety Requirements

Many procedures were established to maintain safety of the working environment [5]. To prevent cross-infection, especially in the intensive care ward and COVID units, it is important to keep isolated imaging and examination rooms, with console and film printers for both X-ray and CT. All non-essential items are removed from these rooms to allow easier, faster and more efficient sterilization. Every time there are suspected or confirmed infections, all equipment must be sanitized and a dedicated pathway to perform radiological examinations must be planned [8,9]. The equipment is properly covered to avoid the cross-contaminations and the floor is covered and the waste always collected after a visit from each patient.

To avoid the spread of nosocomial infection, the contaminated area, semi-contaminated area, and COVID-free area need to be strictly separated and disinfected. If a dedicated examination room cannot be separated from others, strict equipment and air disinfection are required after each patient visit [2].

### Other Safety Procedures Involved Medical Staff in the Radiology Department

Every radiology department should have a designated supervisor for directing and supervising the disinfection and protection of the ward. The supervisor of nosocomial infections must guide the disinfections, make clear divisions and report in time to avoid the infection of staff and patients [5].

Radiographers dedicated to bedside X-ray photography, DR, and CT examinations working in the isolation area are recommended to complete a first period of 14 workdays in the isolation area. Then, they complete a second period of 14 days in a specific dedicated isolation ward for supervised medical observation before returning to normal work due to the possibility of close contact with the confirmed COVID patients [10]. 

Specific radiographers need to be designated and used only when bedside radiography is required. The technicians and the nursing staff have to know and respect the rules of a second level of protection for contact and droplets. Only in the case of splashing secretions of the patient during the procedure is a third level of protection needed. For radiographers, if they enter the room of the exams, they should respect second level protection, otherwise they can respect primary level protection as well as registration staff and all the people that work in clean areas.

Specific COVID rooms have to be identified when a patience is transported and, if it is not possible, a complete sanitization must be performed after each exam. First of all, all unnecessary items must be removed. The rest of the radiology equipment, such as monitors, keyboards and the remaining tools, have to be covered with water resistant materials. Transparent adhesive film is very useful; it allows item use and protection as well. Uncovered surfaces have to pass through cleaning and disinfection procedures [8,9,10].

## 5. Personal Protective Devices

In case an infection of COVID-19 is suspected in the department, staff members should wear the following personal protective equipment (PPE): disposable surgical masks, disposable surgical caps, eye protection (goggles or visor), protective clothing or insulating outer gown, disposable gloves and disposable shoe covers.

At the end of the sanitization procedure, it is essential to wash hands and to rigorously apply a sanitizing gel [2]. 

In case of a confirmed infection from COVID-19 in the department, staff members are required to wear the same personal protective devices previously mentioned, plus N95/99 or FFP2/3 in case of aerosol-generating procedures performed on COVID-19 patients [2]. 

Therefore, in every radiology department, healthcare workers must use surgical masks, as well as eye protection, gloves and gowns; aprons should also be used if gowns are not fluid resistant. Additionally, patients should wear a surgical mask during all imaging examinations. If the patient is to be transported to the radiology department for the examination, the patient must also wear a surgical mask during transport to and from the ward. It is also recommended that radiographers keep a distance of at least 1 m from suspected or confirmed infected patients every time it is possible. The staff must be skilled in dressing and undressing procedures, particularly on how to properly wear, remove, and dispose of a medical mask. The sequence to follow when wearing personal protective equipment is the following: the radiologist first puts on the first surgical cap, then the respirator, then the second surgical cap, then isolation gowns, then the first pair of surgical gloves, the first shoe covers, the protective glass, the disposable gowns, the second surgical gloves, then second shoe covers, then the surgical mask, and at last they must check to the tightness of the outfit. This process takes about half an hour. Then, the medical staff member goes through the two buffer rooms, the potentially contaminated area, and then enters the contaminated area. Then, a flowchart illustrates that the protective equipment must be taken off in different areas when the radiologist has finished carrying out CT scans [2]. After leaving the contaminated area and entering the potentially contaminated area, medical staff first disinfect their hands and then remove their surgical masks, outer shoe covers, outer gloves, disposable gowns, and protective glasses in this strict order. Before each step, it is important to perform hand sanitization and the personal protective equipment must be thrown in a dedicated yellow medical trash can. After this, the medical personnel enter the second buffer room where the inner shoe covers, isolation gown, inner gloves, and outer cap should be taken off. Again, the radiological technician is required to perform hand hygiene before each step. Then they enter the first buffer room, take off the respirator, the inner cap and then put on a surgical mask. Hand hygiene should be strictly followed before each step. It is very important to pay attention so as not to touch the front of the respirator when removing it. Finally, personal cleaning must be performed in the clean area, using a solution of 75% alcohol to disinfect the external auditory canal, iodine to disinfect the nasal cavity and normal saline to disinfect the mouth, and then take a thorough shower for at least 30 min when home [5,7].

## 6. Cleaning and Disinfection

New measures were adopted regarding cleaning and disinfection of the environment and equipment. Regarding environmental daily cleaning, on metal surfaces we use soft detergent and then dry with a dry towel. Detergents that can cause corrosion are strictly not allowed in the normal daily cleaning. Different for plastic surfaces where a simple cleaning with water and common soap, as well for glass that could be cleaned with common glass cleaner, except for amide products [8].

During floor disinfection, visible polluting elements are removed using absorption material at first, and then the floor should be disinfected with chlorinated disinfectant (except for chlorhexidine). Furthermore, doors, windows and handrails should also be disinfected using the same disinfectant [9,10].

Regarding equipment disinfection, after each procedure, the equipment must be disinfected by wiping the surface using 75% ethanol. Corrosive disinfectants are not allowed. Disinfectant sprays should be used with moderation as they may corrode metals and penetrate surfaces, resulting in short-circuit, or other damage. When we need to procced with deep disinfection, the equipment must be shut down, cooled, and completely protected with plastic film the spray for disinfection must not reach radiology equipment that should be cleaned just wiping the surface using 75% ethanol [10]. 

Air disinfection is recommended for the equipment rooms. Ultraviolet irradiation (continuous irradiation for more than 30 min) followed by ventilation for more than 30 min is performed. 

Generally, it is recommended to use non-reusable protections if possible. The reusable protective products, such as protection glasses or visors, should be placed at the disinfection location and soaked in standard chlorine solution (1000 mg/L) or 75% ethanol solutions for more than one hour [8,9,10]. 

Concerning medical waste in the radiology department, all waste from confirmed patients must be managed as contaminated items that should not leave the contaminated without following strict rules. Infectious waste must be put in a medical waste collection bag (ideally not entirely full). The bag has to be sprayed with 5000 mg/L chlorine disinfectant. Then, the inner layer and outer layer must be sealed in a Goose-Neck type and sprayed again with 5000 mg/L chlorine disinfectant. Special identification information must be pasted on the outer layer of the bag and store it in the specialized area for medical wastes. Due to the proximity with contaminated waste, cleaners should protect themselves as if they were in direct contact with the patient. Thus, a secondary level of protection is required [5].

## 7. Healthcare Staff Exposure Monitoring

This organization made it possible to separate the two areas, and it made safety possible by limiting contacts between infected patient and non-infected patients, between infected patients and healthcare staff, and also by limiting contacts within the healthcare staff. Despite all the control measures implemented to contain the healthcare staff contagion, it is impossible to avoid all contact with infected patients or other infected operators. Therefore, strict monitoring is established to find all the risk contacts by the Occupational Medicine of the hospital. Once the risk contacts are identified, the Occupational Medicine plans and carries out the molecular swab test. The operator waiting for the result is placed in isolation to prevent virus propagation.

## 8. Radiology Residents and Medical Students

Due to the need for social distancing in-person conferences, in-person lessons, in-person university exams and didactics have been cancelled in many universities. Many students attending our university programs attend virtual lessons, virtual exams and virtual conferences.

Reducing infections in healthcare workers is critical, so in line with the importance of social distancing medical students were strictly prohibited to attend internships in the hospital wards.

Additionally, most programs reported decreased resident staff, so radiology resident education has been affected by the COVID-19. At the beginning of the outbreak stage in March–April, two resident groups were formed that alternated in the Radiology Department to conduct work shifts over two weeks to limit contacts with the resident staff. As a consequence of this, residents spent less time in hospital, which reduces the benefit of live clinical education.

## 9. Conclusions

The SARS-CoV-2 disease is extremely challenging for every radiology department. The protection of the staff and public must be guaranteed. An organized and coordinated organization is required for a radiology department to ensure all necessary exams and reduce infection risk.

It is crucial to recognize the radiology department as a “frontline” service, essential to the effective functioning of the whole hospital. It is now widely accepted that CT scans play a major role in the diagnosis of COVID-19, and they are considered a tool in early identification and separation of patients presenting potential coronavirus symptoms. Different ways of managing patients must be organized, with specific schedules and isolation measures.

Technicians and physicians should master proper protection and disinfection behaviors and follow procedures as they are both at the frontline in the managing of suspected COVID patients. By respecting these rules, the maximum protection possible can be ensured against new infections and, above all, reduce the stress on medical staff. COVID-19 has an enormous impact on the functioning of the radiology department, driving change in procedures and working style. The measures introduced in this article offer an adequate way to manage these changes. Further articles should focus on how to maintain the educational tasks of a university radiology department despite the pandemic.

### Future Perspectives

The authors are preparing a survey to evaluate the feasibility and the disadvantages/advantages of the implemented actions to face the pandemic situations. The survey will be sent to all the personnel involved in the Radiology Department, also demanding suggestions to improve the department.

The authors suggest also the introduction of technologies tools, based on the e-Health concept, to improve the performances of the Radiology Department.

e-Health refers to the use of tools based on information and communication technologies to support and promote the prevention, diagnosis, treatment, and monitoring of disease and the management of health and lifestyle.

The potential improvements that can be introduced are:

Electronic health record: An electronic health record (EHR), or electronic medical record, refers to the systematized collection of patient and population electronically stored health information in a digital format. These records can be shared across different healthcare settings. Records are shared through network-connected, enterprise-wide information systems, or other information networks and exchanges. EHRs may include a range of data, including demographics, medical history, medication and allergies, immunization status, lab results, diagnostic images, vital signs, detection of anthropometric data, and billing information. 

Telemedicine: Telemedicine is the use of telecommunication and information technology to provide remote medical care. It helps us to eliminate distance barriers and can improve access to medical services that would often not be consistently available in distant rural communities. It is also used to save lives in critical care and emergency situations. Although there were distant precursors to telemedicine, it is essentially a product of the twentieth century telecommunication and information technologies. These technologies permit communications between patient and medical staff with both convenience and fidelity, as well as the transmission of medical, imaging, and health informatics data from one site to another. Actually, in Italy, the Minister of Health has prepared telemedicine guidelines. The authors want to propose this particular branch of e-Health for the Radiology Department as it can be used for the following services:

Providing a consultation with a patient or a specialist assisting the primary care physician in rendering a diagnosis. This may involve the use of live interactive video or the use of store and forward transmission of diagnostic images, vital signs, and/or video clips along with patient data for later review.Remote patient monitoring, including home telehealth, uses devices to remotely collect and send data to a home-health agency or a remote diagnostic testing facility for interpretation. Such applications might include a specific vital sign, such as blood glucose or heart ECG or a variety of indicators for homebound patients. The following services can be used to supplement the use of visiting nurses.

Consumer medical and health information includes the use of the Internet and wireless devices for consumers to obtain specialized health information and online discussion groups to provide peer-to-peer support.

Medical education provides continuing medical education credits for health professionals and special medical education seminars for targeted groups in remote locations.

Consumer Health Informatics: Consumer Health Informatics (CHI) is a subbranch of health informatics that helps us to bridge the gap between patients and health resources. The American Medical Informatics Association defines it as “the field devoted to informatics from multiple consumer or patient views.” The Consumer Health Informatics Working Group of the International Medical Informatics Association defines it as “the use of modern computers and telecommunications to support consumers in obtaining information, analyzing unique health care needs and helping them make decisions about their own health.” CHI includes patient-focused informatics, health literacy, and consumer education. The focus of this field is to allow consumers to manage their own health, using Internet-based strategies and resources with consumer-friendly language. Currently, CHI stands at a crossroads between various healthcare-related fields such as nursing, public health, health promotion, and health education.

Virtual Healthcare teams: Virtual Healthcare teams (VHTs) are groups of health experts cooperating and digitally sharing information about patients. The creation of VHTs would be fundamental in big medical structures such as the Hospital “Policlicnico Tor Vergata” to create good communications between colleagues and between operator of the hospital and the family doctors.

The acquired data coming from the eHealth application can allow the development of software, based on self-learning codes, to be able to give a preliminary evaluation of clinical risk, needs of hospitalization, and social assistance to support the following:

The regional health records;The health system;The research and teaching activities of University of Rome Tor Vergata.

## Figures and Tables

**Table 1 ijerph-18-05255-t001:** Staff members of the Radiology Department of the “Policlicnico Tor Vergata” Hospital who tested positive for SARS-COV2.

Total Number of Radiologists in the Department (Updated to April 2021)	Numbers of Radiologist Tested Positive for SARS-COV2 (2020–2021)	Duration of RT-PCR Positivity per Radiologist
33	5	R.1. From mid-October 2020 to mid-November 2020
Percentage of radiologists that tested positive for SARS-COV2 (2020–2021): 15.15%		R.2. From mid-October 2020 to mid-November 2020
		R.3. From the end of October 2020 to the end of November 2020
		R.4. From the beginning of November 2020 to the end of November 2020
		R.5. From the end of December 2020 to the end of January 2021
Total number of Resident Doctors in the Department (updated to April 2021)	Numbers of Resident Doctors tested positive for SARS-COV2 (2020–2021)	Duration of RT-PCR positivity per Resident Doctor
60	4	R.D.1. From the end of October 2020 to mid-November 2020
Percentage of resident doctors that tested positive for SARS-COV2 (2020–2021): 10.41%		R.D.2. From the end of October 2020 to mid-November 2020
		R.D.3. From the end of October 2020 to mid-November 2020
		R.D.4. From the end of October 2020 to the end of November 2020
		R.D.5. From the end of November 2020 to mid-December 2020
Total number of Technicians in the Department (updated to April 2021)	Numbers of Technicians tested positive for SARS-COV2 (2020–2021)	Duration of RT-PCR positivity per Technician
48	5	T.1. From mid-November 2020 to the beginning of December 2020
Percentage of technicians that tested positive for SARS-COV2 (2020–2021): 6.67%		T.2. From mid-November 2020 to the beginning of December 2020
		T.3. From the end of December 2020 to the end of January 2021
		T.4. From the end of December 2020 to the end of January 2021
Total number of Nurses in the Department (updated to April 2021)	Numbers of Nurses tested positive for SARS-COV2 (2020–2021)	Duration of RT-PCR positivity per Nurse
13	5	N.1. From the beginning of October 2020 to the end of October 2020
Percentage of nurses that tested positive for SARS-COV2 (2020–2021): 38.46%		N.2. From the beginning of November 2020 to the end of November 2020
		N.3. From the beginning of November 2020 to the end of November 2020
		N.4. From the beginning of November 2020 to the end of November 2020
		N.5. From mid-January 2021 to the beginning of February 2021
Total number of Administrative Staff in the Department (updated to April 2021)	Numbers of Administrative tested positive for SARS-COV2 (2020–2021)	Duration of RT-PCR positivity per Administrative Staff Member
12	1	A.S.1. From the end of December 2020 to the end of January 2021
Percentage of administrative staff that tested positive for SARS-COV2 (2020–2021): **8.33%**		

RT-PCR = Real Time − Polymerase Chain Reaction.

## Data Availability

Data sharing not applicable.

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
