# Peer review of "Response of a Radiology Department to the SARS-CoV-2 Pandemic: The Experience of the Hospital “Policlinico Tor Vergata” in Rome"

_ijerph, 2021, doi:10.3390/ijerph18105255_

Round 1
Reviewer 1 Report
Thank you for sharing this paper; it provides an interesting overview detailing the changes to services implemented in an Italian hospital during the COVID-19 pandemic. Given the extreme nature of the pandemic in Italy and the high stress on the health service, a detailed overview of hospital safety responses is of interest both to scholars and health practitioners in the public health and infectious disease fields. However, I am not convinced of this paper’s effective contribution to this literature on COVID-19 service changes, as it is purely descriptive without any attempt to evaluate the reception or effectiveness of these changes. This means that while it lists possibilities for practitioners to implement in similar situations, it gives no insight into whether or not these were successful or whether they had other secondary effects (positive or negative) so that the practitioners looking to the paper for guidance are not fully informed about the outcomes.
I think there is certainly some merit in the details and information the authors have collected: I would suggest building on this initial overview (which is quite short) by conducting some qualitative analysis (for example interviews, focus groups or a short survey) with healthcare workers within the hospital to see their attitudes towards the changes, how they feel it has affected their work and safety. With a robust critical analysis of the outlined changes, the authors could then discuss and make recommendations for what changes were the most useful, and what changes needed further support to effectively implement; this would be a much more powerful contribution with strong policy and practice implications, which the current paper does not make.
I would recommend the authors take this paper back to their research team and use it to plan a further stage of evaluative research, and then resubmit a longer paper at a later stage after conducting and analysing further research evaluating the response.
Author Response
Dear Reviewer,
We have added all the answers in the attached pdf file.
Thank you.

Reviewer 2 Report
- Since this is the experience of your department, it will be interesting to include a table of radiologists, nurses, technicians etc who despite the measures got infected. How did this affect the normal function of the radiology department and how adaptable was it to the temporary (hopefully) loss of staff due to infection. Were there any perturbations in the quality of healthcare provided?
- Since this is a narrative article it will also be useful to visualize by a figure or visual abstract, the radiology department changes between the first and the second SARS-CoV-2 wave. This will also make your work gain more visibility. It will also show, lessons learned, and limitations of the measures taken during the first wave.
- Lastly, another interesting point to be added, is tele-radiology, since “Policlinico Tor Vergata” is a University Medical Centre. The literature is rather scarce on this subject, therefore it will strengthen this work. Either from your current experience from colleagues from other hospitals (especially rural areas) asking for expert radiologist opinion, or as a future perspective. Especially when you mention that students' teaching is disrupted. They will probably (in the near future) need some expert opinion in interpreting their radiological findings. The COVID-19 pandemic is probably here to stay and investing in public health preparedness with insight is a wise public health policy.
- Please proof-read the manuscript and correct your English. For example: P166: patient
Author Response

(The authors gave the same response as above.)

Round 2
Reviewer 1 Report
Thank you for the opportunity to see this work again. It is pleasing to hear that an evaluative future phase will take place regarding staff attitudes to these changes, as that sense of evaluation is the strongest missing piece of the paper. I strongly recommend that this is mentioned before the conclusions in a Strengths and Limitations section: discuss the strengths of this paper (a detailed, practical and implementable outline of service changes) and the limitations (that they have not been evaluated and may not be perfectly replicable) and how further work will be done to address these.
The addition of the measurement of staff infection rates is good, as it is a small step towards some sort of evaluation of the efficacy of the measures/changes. Some of the wording used is unclear - when you say the impact is "non-relevant", is that a quantitative measure (i.e. you have tested for relevance/significance) or a qualitative measure (i.e. you have judged that the level of sickness is not large enough to be a concern)? If the first, readers need to see the mechanics of how this has been tested; if the second, this would seem to be a value judgement by the authors that is not really based in any real-world understanding of the impact on the team/department, and is not really scientifically sound. It would perhaps be better just to present the tables of infection rates, and state that while you cannot directly claim causality (as you have not really statistically tested this) could you perhaps compare your staff's positivity rates to global reported rates and propose that the service changes/interventions might have played a part? (eg , et al. Infection and mortality of healthcare workers worldwide from COVID-19: a systematic review.
The addition of the images is good visual data for readers; however, for accessibility purposes (as well as clarity for general readers), it is important that the visuals presented in all images are fully explained in the text or captions to images.
Author Response
Dear Reviewer good morning,
attached the answer to your comments.
Best Regards,
Dr. Gian Marco Ludovici

Reviewer 2 Report
Dear Editor,
Dear Authors,
I believe you have done excellent work!
Wishing you all the best on citations!
Author Response

(The authors gave the same response as above.)

Round 3
Reviewer 1 Report
The authors have now addressed all reviewer comments, and the final editorial decision is ready to be made. I believe this paper should proceed to publication.